# EXOC6 (Exocyst Complex Component 6) Is Associated with the Risk of Type 2 Diabetes and Pancreatic β-Cell Dysfunction

**DOI:** 10.3390/biology11030388

**Published:** 2022-03-01

**Authors:** Nabil Sulaiman, Mahmood Yaseen Hachim, Anila Khalique, Abdul Khader Mohammed, Saba Al Heialy, Jalal Taneera

**Affiliations:** 1Department of Family Medicine, College of Medicine, University of Sharjah, Sharjah P.O. Box 27272, United Arab Emirates; nsulaiman@sharjah.ac.ae; 2College of Medicine, Mohammed Bin Rashid University of Medicine and Health Sciences, Dubai P.O. Box 505055, United Arab Emirates; mahmood.almashhadani@mbru.ac.ae (M.Y.H.); saba.alheialy@mbru.ac.ae (S.A.H.); 3Sharjah Institute for Medical Research, University of Sharjah, Sharjah P.O. Box 27272, United Arab Emirates; aabid@sharjah.ac.ae (A.K.); amohammed@sharjah.ac.ae (A.K.M.); 4Department of Basic Sciences, College of Medicine, University of Sharjah, Sharjah P.O. Box 27272, United Arab Emirates

**Keywords:** EXCO6, INS-1 cells, insulin secretion, RNA sequencing, siRNA silencing, type 2 diabetes

## Abstract

**Simple Summary:**

EXOC6 and EXOC6B (EXOC6/6B) are components of the exocyst complex involved in the secretory granule docking. The exact role of EXOC6/6B in pancreatic β-cell function and risk of type 2 diabetes (T2D) required further investigations. Herein, we showed that EXOC6/6B is associated with the risk of T2D and is essential to maintain the function of pancreatic β-cell. Our study suggests that EXOC6/6B are crucial regulators for insulin secretion and for exocytosis machinery in β-cells.

**Abstract:**

EXOC6 and EXOC6B (EXOC6/6B) components of the exocyst complex are involved in the secretory granule docking. Recently, EXOC6/6B were anticipated as a molecular link between dysfunctional pancreatic islets and ciliated lung epithelium, making diabetic patients more prone to severe SARS-CoV-2 complications. However, the exact role of EXOC6/6B in pancreatic β-cell function and risk of T2D is not fully understood. Herein, microarray and RNA-sequencing (RNA-seq) expression data demonstrated the expression of EXOC6/6B in human pancreatic islets. Expression of EXOC6/6B was not affected by diabetes status. Exploration of the using the translational human pancreatic islet genotype tissue-expression resource portal (TIGER) revealed three genetic variants (rs947591, rs2488071 and rs2488073) in the EXOC6 gene that were associated (*p* < 2.5 × 10^−20^) with the risk of T2D. Exoc6/6b silencing in rat pancreatic β-cells (INS1-832/13) impaired insulin secretion, insulin content, exocytosis machinery and glucose uptake without cytotoxic effect. A significant decrease in the expression Ins1, Ins1, Pdx1, Glut2 and Vamp2 was observed in Exoc6/6b-silenced cells at the mRNA and protein levels. However, NeuroD1, Gck and InsR were not influenced compared to the negative control. In conclusion, our data propose that EXOC6/6B are crucial regulators for insulin secretion and exocytosis machinery in β-cells. This study identified several genetic variants in EXOC6 associated with the risk of T2D. Therefore, EXOC6/6B could provide a new potential target for therapy development or early biomarkers for T2D.

## 1. Introduction

Diabetes mellitus is a major health problem, diagnosed in more than 500 million individuals worldwide (www.idf.org, accessed on 6 January 2022). Type 2 diabetes (T2D) accounts for approximately 90% of all diabetes cases described by elevated blood sugar levels due to inappropriate insulin secretion levels in pancreatic β-cells, action in target tissues, or both [1].

Insulin is a 51-amino acid protein synthesized specifically in pancreatic β-cells and is governed by different molecules, including sugar, Ca^2+^, ATP, phospholipids, cAMP and hormones [2,3,4,5]. Classically, elevated ATP concentration closes the sensitive K^+^-channels in response to glucose stimulation [6] and opens the voltage-gated Ca^2+^ channels to permit Ca^2+^ influx and trigger the exocytosis of insulin granules. Exocytosis machinery of insulin granules is a well-organized process involving tethering, docking, and fusion of granules to a specific site on the plasma membrane [7]. However, despite the increased knowledge about the exocytosis of insulin granules, genes and molecules involved in the mechanism remain to be discovered. 

The exocyst complex was first identified in the yeast Saccharomyces cerevisiae [8]. Mammalian exocyst complex consists of eight subunits, including Sec3 (Exoc1), Sec5 (Exoc2), Sec6 (Exoc3), Sec8 (Exoc4), Sec10 (Exoc5), Sec15 (Exoc6), Exo70 (Exoc7) and Exo84 (Exoc8), which are anticipated in tethering secretory vesicles to the plasma membrane [9,10,11,12]. In addition, they can regulate different physiological mechanisms, such as cell migration, morphogenesis, cell cycle progression and tumor invasion [13,14,15,16,17]. Furthermore, ablation of exocyst function has been shown to associate with normal secretory granule delivery to exocytic sites, whereas the fusion process was reported to be defected [18].

Numerous studies have reported a role for mammalian exocyst complex in glucose-stimulated insulin secretion or docking mechanism required for granules exocytosis in pancreatic β-cells [19,20,21,22,23,24]. Exoc2, 3, 4 and Exoc5 subunits were shown to express in pancreatic β-cells [19]. Overexpression of dominant-negative mutants of Exoc3, 4, or 5 showed a decrease in the number of vesicles at the plasma membrane or inhibition of insulin secretion [19]. Sec15 (Exoc6) was reported to be an essential component of a multiprotein complex required for exocytosis [25,26].

Mammalian cells contain two genes homologous to Sec15, Exoc6 and Exoc6b [27]. Several studies have reported that Exoc6 participates in insulin signaling and the exocytic machinery in GLUT4 translocation in 3T3-L1 adipocytes [12,28]. Recently, we proposed that EXOC6/6B genes might be an important molecular link between dysfunctional pancreatic islets and ciliated lung epithelium that makes diabetic patients more susceptible to severe SARS-CoV-2 complications [29]. We showed that EXOC6/6B are expressed in the lung and pancreatic tissues as shared genes between the two tissues.

In this study, we aimed to explicitly investigate the role of EXOC6/6b in pancreatic islets β-cell function and insulin secretion. Microarray and RNA-sequencing data were used to profile the expression of EXOC6/6b in human pancreatic islets with/without diabetes. Using the TIGER portal, we also explored whether EXOC6/6b contain genetic variants associated with T2D. Additionally, numerous functional experiments were carried out in INS-1 (832/13) cells, including siRNA silencing, insulin secretion, glucose uptake, reactive oxygen species (ROS) production, cell viability and apoptosis to study the impact of Exoc6/6b in β-cell function. 

## 2. Materials and Methods

### 2.1. Expression Data from Human Pancreatic Islets

The microarray and RNA-seq expression data from isolated human pancreatic islets were obtained from the National Center for Biotechnology Information (NCBI) Gene Expression Omnibus (GEO) publicly available database (accession numbers: GSE50398 and GSE41762) [30,31].

### 2.2. Screening for T2D-Associated Genetic Variants in Exoc6/6b

The TIGER portal (http://tiger.bsc.es, accessed on 6 January 2022) [32] was utilized to explore a region spanning ±100 kb up-and downstream of EXOC6 and EXOC6B individually for the presence of T2D-associated genetic variants. The search was performed in 3 different datasets: the 70K T2D project, DIAGRAM 1000G and DIAGRAM Diamante T2D GWAS meta-analysis.

### 2.3. Maintaining of INS-1 (832/13) Cells 

As described previously, the clonal rat pancreatic β-cells (INS-1 832/13) (kindly provided from Dr. C. Newgard, Duke University, NC, USA) were cultured in complete RPMI 1640 medium, as previously described [33,34].

### 2.4. siRNA Silencing and Insulin Secretion 

The INS-1 cells were transfected as previously described [35] using lipofectamine transfection reagent and siRNA against Exoc6 (s132217) or Exoc6b (500233-SMARTpool) in addition to siRNA negative control. After 48 h post-transfection, a glucose-stimulated insulin secretion assay was investigated. Briefly, silenced and control cells were washed with secretion assay buffer (SAB) [35] containing 2.8 mM glucose. Next, cells were incubated for 2 h in a 2 mL SAB at 37 °C. Insulin secretion measurement was performed by stimulating the cells in 1 mL SAB containing either 2.8 mM glucose, 16.7 mM glucose, 2.8 mM glucose plus 35 mM KCL, or 16.7 mM glucose plus 35 mM KCL for 1 h. A 100 μL of the supernatant was aspirated and used to measure insulin release using a rat insulin ELISA kit (Mercodia, Sweden). Total protein was extracted by mammalian protein extraction reagent (MPER) to determine insulin content and concentration by standard BCA protein assay (Thermo Fisher). Extracted protein was diluted at 1:250; insulin content was measured with a rat insulin ELISA kit (Mercodia) and normalized to the total protein amount.

### 2.5. Cell Viability Assay 

For cell viability and apoptosis analysis, 2 × 10^4^ INS-1 cells in 100 μL RPMI media were cultured in a 96-well plate for 24 h and transfected as previously described earlier. After 48 h of transfection, 10 μL of MTT (3-(4,5-Dimethylthiazol-2-yl)-2,5-Diphenyltetrazolium Bromide, Sigma, St. Louis, MI, USA) was added into each well and incubated for 3–4 h at 37 °C. Afterward, MTT formazan product was dissolved in dimethyl sulfoxide (DMSO) and absorbance was read by microplate reader at 570 nm (Crocodile mini workstation; Crocodile Control Software). The average absorbance was used to measure the percentage (%) of cell viability using this formula: % cell viability = (OD 570 nm of sample/OD 570 nm of control) × 100. 

### 2.6. Apoptosis Assay

Transfected INS-1 cells in 24 well-plate were washed and re-suspended in 500 μL of Annexin-V Binding Buffer (Becton Dickinson (BD) Biosciences, Franklin, NJ, USA). Annexin V-FITC and propidium iodide (5 μL each) were incubated with the cells for 10 min, then analyzed using BD FACS Aria III flow cytometer (BD, USA).

### 2.7. Glucose Uptake Assay

The 2-NBDG glucose uptake assay kit (#N13195, Invitrogen, Waltham, MA, USA) was used to evaluate glucose uptake following the manufacturer’s guidelines. Briefly, the 2-NBDG reagent was added to the transfected cells and incubated for one hour. Then, cells were washed with PBS and analyzed immediately with flow cytometry using the FITC detector (Excitation/Emission 485/535 nm).

### 2.8. Intracellular Reactive Oxygen Species (ROS)

ROS measurements was performed by the H_2_O_2_ assay, as previously described [36]. In brief, 2 × 10^4^ of cells/well in a total volume of 80 µL were seeded in a 96-wells plate. After 48 h of post-transfection, a 20 µL of the H_2_O_2_ substrate solution was added and the plate was incubated at 37 °C for 4 h. Next, 100 µL of the ROS-Glo detection solution was added to each well at the end of the incubation and incubated at room temperature for 20 min. Immediately, the luminescence was recorded using a plate reader. The average values were used to calculate the relative luminescence unit (RLU).

### 2.9. Quantitative PCR (qPCR)

The cDNA synthesis was generated using a high-capacity cDNA reverse transcription kit (Thermo Fisher, Waltham, MA, USA). Expression of target genes were assessed by using TaqMan gene expression assays; Taqman: EXOC6 (Rn00570613_m1), EXOC6b (Rn01420854_m1), Glut2 (Rn00563565_m1), Ins1 (Rn02121433_g1), Ins2 (Rn01774648_g1), Pdx1 (Rn00755591_m1), Insr (Rn00690703_m1), Gck (Rn00561265_m1) and Rat Hprt1 (Rn01527840_m1) as endogenous (all from Thermo Fisher). SYBR green gene expression analysis was performed with the corresponding primers (Table 1). The qPCR reactions were conducted in triplicate in 96-well plates using the QuantStudio 3 qPCR system (Applied Biosystems, Waltham, MA, USA). Relative gene expression was determined by the 2^−ΔΔCT^ method.

### 2.10. Western Blot Analysis 

Western blot expression analysis was performed as previously described [36]. Briefly, transfected cells were collected in ice-cold PBS. Samples were centrifuged at 1200 rpm for 10 min at 4 °C; the cell pellet was resuspended in mammalian protein extraction reagent (M-PER) containing 1% protease cocktail inhibitor and 1% phenylmethylsulfonyl fluoride (PMSF). The samples were thoroughly vortexed and kept on ice for 30 min and then centrifuged at 12,000 rpm for 15 min at 4 °C. The supernatant was collected, and protein concentrations were determined by BCA protein assay kit (Thermo Fisher). A 30µg protein was used for Western blot. The blot was incubated overnight at 4 °C with primary antibodies against INSULIN (Cat. #8138-anti-mouse, dilution 1:1000), INSR-β (Cat. #23413-anti-rabbit, dilution 1:1000), and VAMP2 (Cat. # 13508-anti-rabbit, dilution 1:1000) purchased from Cell Signaling Technology (Danvers, MA, USA); PDX1 (Cat. #ab47267-anti-rabbit, dilution 1:1000), NEUROD1 (Cat. #ab213725-anti-rabbit, dilution 1:1000), GCK (Cat. #ab37796-anti-rabbit, dilution 1:1000) from Abcam (Cambridge, UK) and GLUT2 (Cat. #A12307-anti-rabbit, dilution 1:1000) from ABclonal (Waltham, MA, USA). The β-actin endogenous control was used from Sigma (Cat. #A5441-anti-mouse, dilution 1:1000). The primary antibodies were incubated at 4 °C overnight. The secondary HRP-linked anti-mouse (CAT. #7076S, dilution 1:1000) and HRP-linked anti-rabbit (#7074S, dilution 1:1000) from Cell Signaling Technology antibodies were used for 1 h. Chemiluminescence was detected using the Clarity ECL substrate kit (Bio-Rad, Berkeley, CA, USA). Protein bands were analyzed using the Bio-Rad Image Lab software (ChemiDoc Touch Gel Western Blot Imaging System; Bio-Rad, Berkeley, CA, USA).

### 2.11. Statistical Analysis 

For differential expression analysis between diabetic and nondiabetic islets, Student’s *t*-test or nonparametric Mann–Whitney U tests were used. For correlation analysis, we used nonparametric Spearman’s test. All statistical analyses were performed using GraphPad Prism (version 8.0.0 for Windows, GraphPad Software, San Diego, CA, USA). Differences were considered significant at *p* < 0.05. 

## 3. Results

### 3.1. Expression of EXOC6 and EXOC6B in Human Pancreatic Islets

Expression of EXOC6/6B in human islets is not well-profiled. Herein, we used our previous microarray and RNA-seq expression data from human islets (publicly available) to investigate the EXOC6/6B expression compared to some key functional genes for human pancreatic β-cell, such as KCNJ11 (potassium inwardly rectifying channel subfamily J member 11) and GLUT1 (solute carrier family 2 member 1; SLC2A1). Microarray expression analysis revealed that EXOC6/6B are expressed in human islets (Figure 1A). Furthermore, expression of EXOC6/6B was shown to have higher expression than KCNJ11 but lower than GLUT1 genes (*p* < 0.05) (Figure 1A). On the other hand, RNA-seq expression analysis showed higher expression of EXOC6/6B relative to KCNJ11 and GLUT1 (*p* < 0.05) (Figure 1B). Moreover, we investigated the impact of diabetes status on the EXOC6/6B expression using RNA-seq data. As shown in Figure 1C, a similar expression of EXOC6/6B was observed in diabetic/hyperglycemic islets (HbA1c ≥ 6%) compared to nondiabetic/normoglycemic (HbA1c < 6%) islets.

### 3.2. Associated Genetic Variants in EXOC6/6B with the Risk of T2D

Using the TIGER portal, we explored whether EXOC6/6B contain any genetic variants (single nucleotide polymorphism, SNP) associated with T2D. At *p* < 0.05, we found 155 SNPs in the 70K for the T2D project, 224 SNPs in the DIAGRAM 1000G and 337 SNPs in DIAGRAM Diamante in the EXOC6 gene. For EXOC6B, 47 155 SNPs in the 70K for T2D project, 227 SNPs in the DIAGRAM 1000G and 131 SNPs in DIAGRAM Diamante were associated with T2D. However, only three SNPs (rs947591, rs2488071 and rs2488073) in DIAGRAM Diamante and 1000G database showed a powerful association with T2D using a cutoff threshold (*p* < 1.0 × 10^−10^) that passed genome-wide significance (Table 2). However, we could not find any expression quantitative trait loci (eQTL) for these SNPs in human islets using the TIGER portal.

### 3.3. Expression Silencing of Exoc6/6B Cells Reduces Insulin Secretion and Influence Cell Functions in INS-1 Cells

To evaluate the impact of Exoc6/6b silencing on insulin secretion, we silenced the expression of each gene individually in INS-1 (832/13) cells using siRNA. Transfection efficiency determined by qPCR expression analysis 48 h post-transfection revealed a significant reduction (~90%; *p* < 0.05) for Exoc6 (Figure 2A) and ~75%; *p* < 0.05 for Exoc6B compared to the negative control (Figure 2B). Next, we investigated the impact of Exoc6/6B-silencing on cell viability and apoptosis levels. As shown in Figure 2C,D, silencing of Exco6/6b on cell viability measured by MTT assay showed no differences between transfected cells and negative control cells. In addition, we could not observe any effect of Exoc6/b silencing on apoptosis as determined by Annexin-V/PI apoptosis analysis compared to negative control cells (Figure 2E,F). The percentage of apoptotic cells in siRNA negative control-silenced cells was 7% compared to 9% in Exoc6-silenced cells. Similarly, the rate of apoptotic cells in siRNA negative control-silenced cells was 5% compared to 6% in Exoc6b-silenced cells. Next, as the exocytosis machinery involves other exocyst complex components (Exoc1-8), we investigated the effect of Exoc6 or Exoc6b silencing of these subunits. As shown in Figure 2D,E, qPCR expression data revealed that only Exoc5 was affected by Exoc6 silencing (*p* < 0.05). No significant effect was observed on the other exocyst components by silencing Exoc6b.

Transfected cells stimulated with glucose exhibited a significant decrease in insulin secretion at 2.8 mM glucose and 16.7 mM glucose in Exoc6/6b-silenced INS-1 cells (*p* < 0.05) (Figure 3A,B). Notably, a significant reduction (~55%; *p* < 0.05) in insulin secretion was observed in Exoc6/6B-silenced INS-1 cells stimulated with 35 mM KCl (a depolarizing agent) compared to control cells, which suggest a defects in the exocytosis machinery (Figure 3A,B). More, measurement of insulin content in Exoc6/6B-silenced cells revealed a significant reduction (~40%; *p* < 0.05) than the negative control cells (Figure 3C). Assessment of ROS production in Exoc6/6b-silenced cells indicated a significant elevation in Exoc6 compared to the negative control (Figure 3D). Exoc6b-silenced cells did not show any change in ROS production levels (Figure 3D). Finally, glucose uptake determination showed a marked reduction (~30%; *p* < 0.05) in Exoc6/6b-silenced cells compared to control cells (Figure 3E).

### 3.4. Silencing of Exoc6/6b Modulates the Expression of genes Involved in β Cell Function

The effect of Exoc6/6b silencing on the pancreatic β-cells functional genes was examined at mRNA and protein levels. mRNA expression of genes involved in insulin production revealed a significant (*p* < 0.05) down-regulation of Ins1, Ins2 and Pdx1, whereas, NeuroD1 was not affected (Figure 4A,B). Likewise, a substantial reduction in protein expression was observed in pro/insulin and PDX1 (*p* < 0.05) relative to control cells. NEUROD1 expression was not affected (Figure 4C,D). Analysis of mRNA and protein expression of genes involved in glucose-sensing (Glut2 and Gck) revealed no difference at mRNA level in Exoc6-silenced cells (Figure 4A), whereas expression of both genes was down-regulated in Exoc6b-silenced cells (Figure 4B). GLUT2 expression was down-regulated at protein level in Exoc6/6b-silenced cells, but Gck was not affected (Figure 4C,D). Expression of InsR, a gene involved in insulin signaling, was shown to be reduced (*p* < 0.05) in Exoc6-silenced cells at mRNA and protein levels (Figure 4A–C). In contrast, mRNA expression of InsR was elevated in Exoc6b-silenced cells but not affected at the protein level (Figure 4B–D). Finally, mRNA expression of Vamp2 was down-regulated in Exoc6/6b-silenced cells (Figure 4A,B); however, it was reduced only in Exoc6-silenced cells at the protein level. (Figure 4C,D).

## 4. Discussion

EXOC6/6B are essential for insulin granule docking/secretion and for insulin signaling through interaction with the glucose transporter GLUT4 [12]. This emphasizes the important role that EXOC6/6B play in the pathophysiology of T2D. This study showed that EXOC6/6B are expressed in human pancreatic islets, but their expression was not influenced by diabetes or hyperglycemia status. Three SNPs (rs947591, rs2488071, and rs2488073) in the proximity of EXOC6 gene were found to be strongly associated with T2D in the DIAGRAM Diamante and DIAGRAM 1000 G GWAS database. We also showed that silencing of Exoc6 or Exoc6b in INS-1 cells impaired insulin secretion, exocytosis machinery, reduced glucose uptake, elevated ROS production, and disturbed several pancreatic β-cell functional genes at the protein and mRNA levels. Together, these results suggest that Exoc6/6b are essential for β-cell biology.

Accumulative data suggest that EXOC6/6B and other members of the exocyst complex have a diverse physiological function that depends on the microenvironment. For example, EXOC6/6B was shown to function as cilia-associated proteins with ciliogenesis-associated roles [37,38]. A process that is essential in developing epithelial organs, including the lung and kidney [39]. Loss of function of Exco6 in erythrocytes (cell with high heme production and iron turnover) resulted in anemias and heme defects [40]. Furthermore, it has been recently demonstrated that EXOC6/6B promotes the malignant transformation of human bronchial epithelial cells (HBECs) [41]. In addition, it was proposed that the exocyst complex is important in the epithelium barrier integrity [42]. In this context, it is well-accepted that exocytosis is the main mechanism used to the secretion of surfactants involved in alveolar host defense in the lungs. Therefore, defect in the exocytosis machinery may implicate surfactant secretion, leading to severe lung disorders [43]. Recently, we reported that SARS-CoV-2 infection modulates the expression of EXOC6/6B in lung tissues. This might shed light on the possible pathophysiological mechanisms shared between T2D and COVID-19 [29].

To the best of our knowledge, this is the first report investigating the expression of EXOC6/6B in human islets and their association with the pathophysiology of diabetes. The finding that Exoc6/6b is not influenced in diabetic/hyperglycemic islets compared to nondiabetic/normoglycemic is not surprising. A possible explanation could be a compensatory response that regulates the expression of EXOC6/6B in β-cell or reflects that the diabetic donors suffer from insulin resistance rather than insulin secretion.

Interestingly, exploration of genetic variants associated with T2D identified three different SNPs (rs947591, rs2488071 and rs2488073) in the proximity of EXOC6 gene. The rs947591 was first identified in relation to T2D risk on chromosome 10, located in IDE (insulin-degrading enzyme), KIF11 (the kinesin interacting factor 11) and HHEX (the hematopoietically expressed homeobox) gene cluster [44]. Similarly, the same SNP was replicated in Chinese Han populations [45]. Moreover, rs947591 was associated with other traits, including triglyceride and total cholesterol levels in the Chinese population [46]. rs2488071 was identified as a novel functional locus shared between T2D and birth weight in Chinese populations [47]. It seems that rs947591 is an important genetic variant that requires replication in different ethnic populations as biomarkers for the risk of T2D. Importantly, exploring the effect of rs947591, rs2488071, and rs2488073 on EXOC6 and EXOC6B expression in human pancreatic islets using TIGER portal revealed no effect. Further investigations are needed to explore effect of the 3 identified SNPs on other tissues such muscle or blood. 

Exco6 was proposed as necessary machinery for the β-cells function by enabling the docking of secretory vesicles to the membrane before the SNARE-mediated fusion (12). However, the specific role of EXOC6/6b in pancreatic β-cells is not clear. Gene silencing of Exoc6/6b in INS-1 cells impaired insulin secretion reduced insulin content and decreased glucose uptake without affecting cell viability or cytotoxic apoptosis levels (Figure 2 and Figure 3). As expected, the observed reduction of insulin secretion in cells stimulated with the depolarizing agent KCl, clearly demonstrated that silencing of Exoc6/6b affects the exocytosis machinery of insulin release. 

Notably, genes involved in insulin biosynthesis, such as Ins1, Ins2 and Pdx1, were down-regulated at mRNA and protein levels in Exoc6/6b-silenced cells. Therefore, it is not clear how Exoc6/6b affected insulin biosynthesis. However, previous reports demonstrated that Rab protein, which belongs to the small GTPase family, regulates insulin synthesis, exocytosis, ER-associated degradation of proinsulin or mediates proinsulin to insulin conversion in β-cells [12,13,48,49]. Considering that Exoc6 directly interacts with Rab protein to participate in GLUT4 translocation in adipocytes [12], it is reasonable to speculate that such interaction between Rab-Exoc6 might be involved in insulin biosynthesis. However, further investigations will be required to address this issue. 

Expression of Glut2 that involved in glucose sensing and uptake machinery was down-regulated in Exoc6/6b-silenced cells. It has been shown that defects in the machinery of glucose-sensing resulted in insulin secretion impairment and severe hyperglycemia [50,51]. This work demonstrated that the decreased Glut2 expression accompanied low glucose uptake efficiency, a crucial driving factor for insulin secretion.

Moreover, the observed expression decreased in InsR, involved in insulin signaling, in Exco6-silenced cells might affect the insulin secretion. Down-regulation of InsR in β-cell has been shown previously to impair insulin secretion [52]. Together, the data indicate that Exoc6/6b are involved in insulin secretion by reducing the expression of glucose sensing and insulin signaling regulators.

In contrast to our data, it has been shown that overexpression of Exoc6 in mouse insulinoma βtc6 cells did not affect insulin secretion [53]. However, double overexpression of EXOC6 and SYTL3 significantly reduced insulin secretion [53]. Such conflicting data might be due to the different cell lines used for validation, the transfection protocol, or the experimental setup. 

## 5. Conclusions

Our data suggest that EXOC6/6B are essential components in the exocytosis machinery and insulin release in β-cell function. In addition, several genetic variants associated with the risk of T2D were found in the EXOC6 gene. Therefore, further works are warranted to explore whether EXOC6/6b can be targeted for potential therapy or biomarkers for T2D.

## Figures and Tables

**Figure 1 biology-11-00388-f001:**
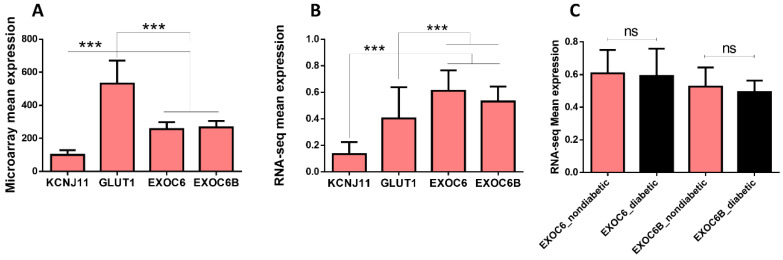
Expression of EXOC6/6b genes in human pancreatic islets. (**A**) Microarray mean expression of KCNJ11, GLUT1, EXOC6 and EXOC6B in nondiabetic human islets (*n* = 45). (**B**) RNA sequencing expression of KCNJ11, GLUT1, EXOC6 and EXOC6B in nondiabetic human islets (*n* = 50). (**C**) Differential expression analysis (RNA-seq) of EXOC6 and EXOC6B in human islets obtained from diabetic/hyperglycemic donors (*n* = 27) compared to nondiabetic/normoglycemic donors (*n* = 50). ***; *p* < 0.001. n.s.; not significant. Bars represent mean  ±  SD.

**Figure 2 biology-11-00388-f002:**
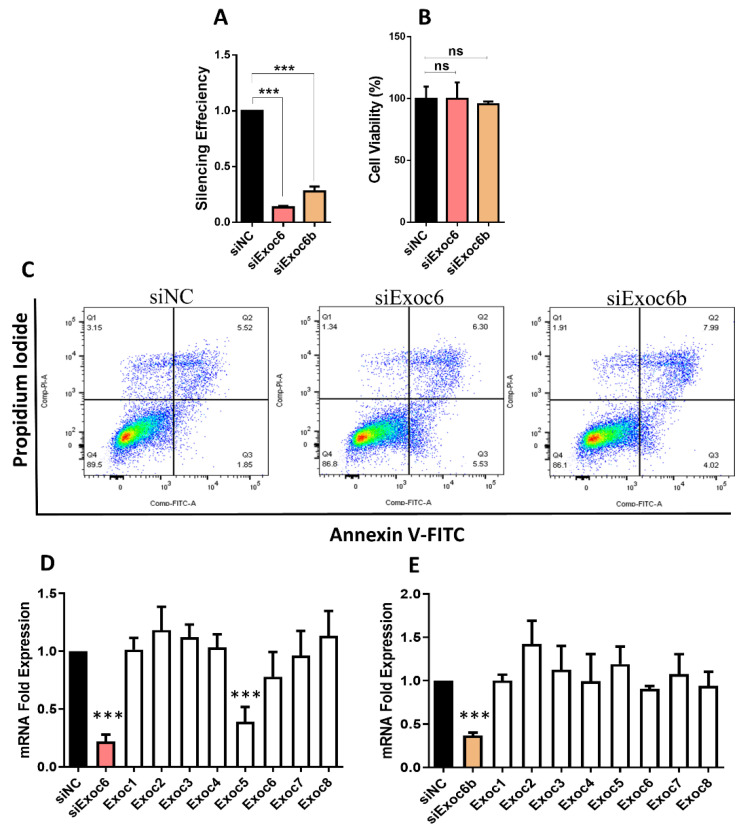
Silencing of Exoc6/6b in INS-1 cells. (**A**) Expression analysis of Exoc6 and Exoc6b as determined by qPCR expression 48 h post-transfection. (**B**) Percentage of cell viability evaluated by MTT assay in Exoc6/6b-silenced cells and control cells. (**C**) Apoptosis level in Exoc6/6b-silenced INS-1 cells compared to negative control cells determined by flow cytometry analysis. (**D**–**E**) qPCR expression analysis of exocyst subunits in Exoc6 (**D**) or Exoc6b-silenced cells (**E**). ***; *p* < 0.001. n.s.; not significant. Bars represent mean ± SD. Data were acquired from 3 different experiments.

**Figure 3 biology-11-00388-f003:**
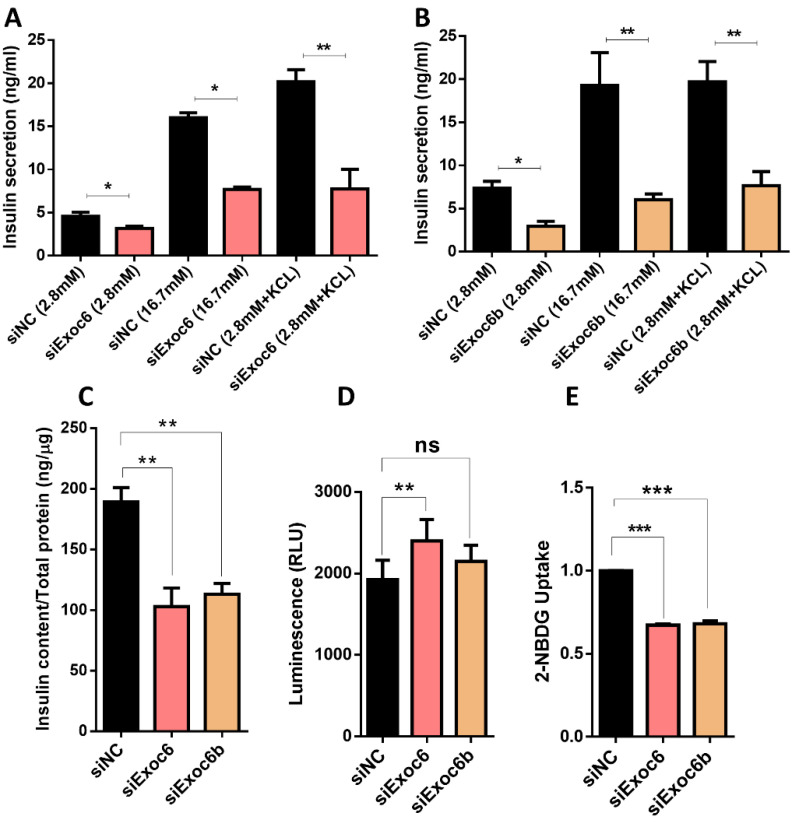
Impact of Exoc6/6b silencing on insulin release in INS-1 cells. (**A**,**B**) Measurement of insulin secretion (normalized to cellular protein content) stimulated with 2.8 mM glucose, 16.7 mM glucose, or 35 mM KCl + 2.8 mM glucose in Exoc6 (**A**) or Exoc6b (**B**) silenced cells compared to control cells. (**C**) Insulin content measurements normalized to protein content in Exoc6/6b-silenced cells compared to control cells. (**D**) assessment of ROS levels by fluorescence intensity in Exoc6/6b-silenced INS-1 cells compared to control cells. (**E**) Determination of glucose uptake in Exoc6/6b-silenced cells compared to control cells. *; *p* < 0.05. **; *p* < 0.01. ***; *p* < 0.001. ns.; not significant. Bars represent mean ± SD. Data are acquired from 3 different experiments.

**Figure 4 biology-11-00388-f004:**
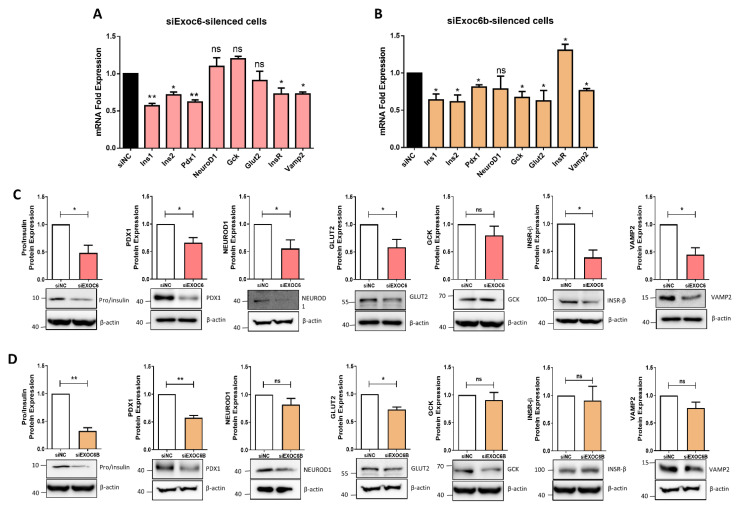
Impact of Exoc6/6b silencing on the expression of β-cell function genes. Total mRNA/protein was extracted from Exoc6/6b-silenced cells or siRNA negative control after 48 h of transfection and subjected for qPCR or western blot analysis. qPCR expression analysis of Ins1, Ins2, Pdx1, NeuroD1, Gck, Glut2, InsR and Vamp2 in *Exoc6-*silenced cells (**A**) or Exoc6b-silenced cells (**B**) compared to control cells. Western blot analysis of Pro/insulin, PDX1, NEUROD1, PDX1, GLUT2, GCK, INSβ and VAMP2 relative to the endogenous control protein β-actin in Exoc6-silenced cells (**C**) or Exoc6b-silenced cells (**D**). Data were obtained from three independent experiments (Appendix A). *; *p* < 0.05. **; *p* < 0.01. n.s.; not significant. Bars represent mean  ±  SD.

**Table 1 biology-11-00388-t001:** SYBR green primers sequences.

S. No.	Genes/Symbol	Forward Primers (5′-3′)	Reverse Primers (5′-3′)
1.	Exoc1	TGTCAAGATGAGCCACCACG	GGCGATGCTCTCAGGTTCAC
2.	Exoc2	ACTCCCTGCAGTCGTTGAAG	CCTGGGTTTTAGGCTGCTGA
3.	Exoc3	CAGCTGCGCGGATGTGTA	GTTGCAACAGCCTCCAGGTC
4.	Exoc4	TGCAAACCTGGAGCCAGAAAT	CGAAGGAGACACTGTTTGGC
5.	Exco5	ACCGAAGGTTCCAAGATGCT	ACATCTCCAACACTGGCAGG
6.	Exoc7	CCATTGGGGCCAAAGCTCTA	AACGGTGCCATCTTTAGGCA
7.	Exoc8	TCGAAGGGCAGTGTCTCAAC	CAATTCGAAGCTGGCGGATG
8.	Hprt	TTGTGTCATCAGCGAAAGTGG	CACAGGACTAGAACGTCTGCT
9.	Vamp2	TGGTGGACATCATGAGGGTG	GCTTGGCTGCACTTGTTTCA

**Table 2 biology-11-00388-t002:** SNPs in the proximity of EXOC6 (±100 Kb) that associated with T2D in TIGER data portal.

DIAGRAM DIAMANTE									
ID	Reference Allele	Alternate Allele	Type	Sample Size	Effect Allele	MAF	OR	SE	*p*-Value
rs947591	C	A	Upstream	231420	A	0.48	1.07	0.0064	2.4 × 10^−25^
rs2488071	A	G	Upstream	231420	A	0.46	1.048	0.0064	9.1 × 10^−21^
rs2488073	A	G	Upstream	231420	A	0.48	1.047	0.0064	2.5 × 10^−20^
DIAGRAM 1000G									
rs947591	C	A	Upstream		A	-	1.09	0.013	7.4 × 10^−13^
rs2488073	A	G	Upstream		A	-	0.917	0.013	8.5 × 10^−12^
rs2488071	A	G	Upstream		A	-	0.919	0.013	3.1 × 10^−11^

## Data Availability

Not applicable.

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
