# Peer review of "EXOC6 (Exocyst Complex Component 6) Is Associated with the Risk of Type 2 Diabetes and Pancreatic β-Cell Dysfunction"

_biology, 2022, doi:10.3390/biology11030388_

Round 1

Reviewer 1 Report

The paper "EXOC6 (exocyst complex component 6) is a novel gene associated with the risk of type 2 diabetes and required for pancreatic 3 β-function" by Sulaiman and colleagues is a reanalysis of previously published transcriptomics data to re-confirm the downregulation of EXOC6 in diabetic tissue. The authors confirm this with further experiments, and add the effect of Exoc6 silencing on the expression of pancreatic genes involved in diabetes.

- The gene EXOC6 is not "novel", in any sense. It is an evolutionary conserved protein involved in vesicle generation and transport across eukaryotes. Also, the involvement of EXOC6 in T2D and insulin secretion is not novel, as it was discovered by Suriben and colleagues in 2015 (https://www.sciencedirect.com/science/article/pii/S0092867415014798). I suggest therefore a change in the title

- It is not true that the role of EXOC6 in pancreatic metabolism and T2D is not understood. A review in 2016 describes the gene involvement in insulin production (https://www.nature.com/articles/nrm.2015.21) and more papers describe its role in diabetes (e.g., https://journals.physiology.org/doi/full/10.1152/ajpendo.00109.2019). So, such claims in the summary, introduction and discussion should be changed.

- The normalization of RNA-Seq data is problematic: comparing different transcripts with different lengths (e.g. in Figure 1B) requires further normalization (such as FPKM).

- In Figure 4D, the axis labels specify siEXOC6, but they should specifiy siExoc6b.

- The beta-actin control of Figure 4C in NEUROD1 (siEXOC6 experiment) looks very similar to the control bands of Figure 4D PDX1, GLUT2 (siEXOC6b experiment) at different exposure. This is not possible, because the two experiments should be different.

- Given the centrality of the siExoc6/siExoc6b experiments in the usefulness of this study, I believe one cell line (INS-1 cells) is not enough to confirm these findings, aggraveted by it being a non-human model. The authors should confirm their experiments in another pancreatic B-cell model (rat or otherwise)

- In the summary, "EXOC6 and EXOC6B (EXOC6/6B) are components of the exocyst complex are 16 involved in the secretory granule docking": "are" si repeated twice

Author Response

Reviewer 1:

The paper "EXOC6 (exocyst complex component 6) is a novel gene associated with the risk of type 2 diabetes and required for pancreatic 3 β-function" by Sulaiman and colleagues is a reanalysis of previously published transcriptomics data to re-confirm the downregulation of EXOC6 in diabetic tissue. The authors confirm this with further experiments, and add the effect of Exoc6 silencing on the expression of pancreatic genes involved in diabetes.

- The gene EXOC6 is not "novel" in any sense. It is an evolutionary conserved protein involved in vesicle generation and transport across eukaryotes. Also, the involvement of EXOC6 in T2D and insulin secretion is not novel, as it was discovered by Suriben and colleagues in 2015 (https://www.sciencedirect.com/science/article/pii/S0092867415014798). I suggest therefore a change in the title.

Answer: Based on the reviewer's suggestion, we changed the title and removed the word novel genes “EXOC6 (exocyst complex component 6) is associated with the risk of type 2 diabetes and pancreatic β-cell dysfunction.”

- It is not true that the role of EXOC6 in pancreatic metabolism and T2D is not understood. A review in 2016 describes the gene involvement in insulin production (https://www.nature.com/articles/nrm.2015.21) and more papers describe its role in diabetes (e.g., https://journals.physiology.org/doi/full/10.1152/ajpendo.00109.2019). So, such claims in the summary, introduction and discussion should be changed.

Answer: We thank the reviewer for the comment. We changed the statement from not understood to required further investigations or not fully understood. We also included the two references in the bibliography.

- The normalization of RNA-Seq data is problematic: comparing different transcripts with different lengths (e.g. in Figure 1B) requires further normalization (such as FPKM).

Answer: The RNA-sequencing were prepared using Illumina's TruSeq RNA Kit and libraries were checked for quality using the 2200 TapeStation System (Agilent Technologies, USA). The output reads were aligned to the human reference genome (hg19) with STAR. Raw data were normalized using trimmed mean of M-values and presented as Fragments Per Kilobase of Exon Per Million Fragments Mapped (FPKM) or transformed into log2 counts per million using the voom-function (edgeR/limma R-packages). Please see our original publication (https://www.ncbi.nlm.nih.gov/pmc/articles/PMC4183326/) where we described the generation and normalization of the RNA-seq.

- In Figure 4D, the axis labels specify siEXOC6, but they should specifiy siExoc6b.

Answer: we apologize for this. The figure has been corrected (see below).

- The beta-actin control of Figure 4C in NEUROD1 (siEXOC6 experiment) looks very similar to the control bands of Figure 4D PDX1, GLUT2 (siEXOC6b experiment) at different exposure. This is not possible, because the two experiments should be different.

Answer: Thank you for your comment. The beta-actin of Figure 4C in NEUROD1 looks similar to the beta-actin of Figure 4D PDX1 and GLUT2, but it is not the same. (Please see the below figure with better magnification).

- Given the centrality of the siExoc6/siExoc6b experiments in the usefulness of this study, I believe one cell line (INS-1 cells) is not enough to confirm these findings, aggraveted by it being a non-human model. The authors should confirm their experiments in another pancreatic B-cell model (rat or otherwise).

Answer: We agree with the reviewer’s comment that using another pancreatic β-cell model will further confirm our data. In this study, we used the INS-1 (831/13) cell line widely used in diabetes research. INS-1 cells have stable human preproinsulin, robust glucose-induced insulin secretory responses, high insulin contents and express glucosidase (https://pubmed.ncbi.nlm.nih.gov/10868964/). Unfortunately, we do have the INS-1 cells in our lab. Therefore, adding a new pancreatic beta cells model and running all/some functional experiments will take a long time, bearing in mind that we have only 10 days for the re-submission.

- In the summary, "EXOC6 and EXOC6B (EXOC6/6B) are components of the exocyst complex are 16 involved in the secretory granule docking": "are" si repeated twice

Answer:  Sorry for that. We corrected it.

Reviewer 2 Report

Exocytosis represents an important biological process, and the exocyst complex, a large, evolutionarily conserved complex consisting of 8 subunits that include Sec15 (EXOC6/EXOC6B), is essential for exocytosis. In this manuscript, the authors hypothesized that EXOC6/6B component (consisting of 2 genes, i.e., EXOC6 gene and EXOC6B gene in animal species, e.g., humans and rats) of the exocyst complex in pancreatic islets (i.e., pancreatic beta-cells) are critical for insulin secretion, and dysregulations of EXOC6 and EXOC6B gene products could be involved in pathophysiology of type 2 diabetes (T2D). The authors presented publicly available microarray expression data from human pancreatic islets to compare human EXOC6 and EXOC6B gene expressions between non-diabetic human pancreatic islets, and diabetic human pancreatic islets, and found no significance difference. Then, the authors searched for selected sets of single nucleotide polymorphisms (i.e., SNPs) located in either EXOC6 gene or EXOC6B gene loci for association with T2D risk in previously conducted human genome-wide association studies (GWASs), and identified 3 SNPs located in EXOC6 gene. i.e., rs947591, rs2488071, and rs2488073, that reached genome-wide significance [i.e., P-value < 5*10^(-8)] for association with T2D based on 2 different online portals: TIGER data portal and T2D knowledge portal, respectively. The authors then used the INS-1 (832/13) cell line model to assess the effects of siRNA silencing of Exoc6 and Exoc6b on insulin secretion, insulin content, reactive oxygen species (ROS) level, and glucose update in INS-1 (832/13) cells, and further evaluated the impacts of silencing of Exoc6 and Exoc6b gene expressions on the expressions of a selected set of pancreatic beta-cell function genes, i.e., Ins1, Ins2, Pdx1, NeuroD1, Gck, Glut2, InsR and Vamp2 at mRNA expression [using quantitative PCR (qPCR)] and protein expression (Western blot analysis) levels, respectively. The results obtained by qPCR and Western blot analysis were not entirely consistent at mRNA level and at protein level: at mRNA level, Gck gene expression was down-regulated in Exoc6b-silenced cells but at protein level, Gck gene expression was not affected, and at mRNA level, InsR gene expression was up-regulated in Exoc6b-silenced cells but at protein level, InsR gene expression was down-regulated, but did not reach statistical significance. Based on the data presented in this manuscript, authors then draw the conclusion that Exoc6/6b could be essential for the β cells function and might be a novel therapeutic target for treatment of T2D.

(I) Major Comments

I have significant concerns that are elaborated in the following.

(1) As indicated in Section 2.1, the authors used 2 Microarray and RNA-sequencing (RNA-seq) data sets from human pancreatic islets obtained from NCBI Gene Expression Omnibus (GEO) database, i.e., GSE50398 [URL: https://www.ncbi.nlm.nih.gov/geo/query/acc.cgi?acc=GSE50398; Platforms: expression profiling by Affymetrix Human Gene 1.0 ST Array and Illumina HiSeq 2000 (Homo sapiens)] and GSE41762 (URL: https://www.ncbi.nlm.nih.gov/geo/query/acc.cgi?acc=GSE41762; Platform: Affymetrix Human Gene 1.0 ST Array), respectively, but the gene expression levels for EXOC6 and EXOC6B genes were not changed in diabetic pancreatic islets compared to non-diabetic pancreatic islets. These RNA gene expression results are very essential, but they do not provide support for potential dysregulations of EXOC6 and EXOC6B genes at the mRNA level between diabetic and non-diabetic pancreatic islet tissues. Further, these gene expression data sets are not original data set generated by the authors, but were publicly available datasets stored in NCBI GEO database, and this severely limited the originality of the study, and the fact that EXOC6 and EXOC6B genes were not differentially expressed in 2 independent microarray and RNA-seq data sets were essentially negative findings, which did not support the authors’ study hypothesis, which is another major limitation of this study.

(2) The authors searched 2 online data portals, i.e., TIGER data portal (URL: http://tiger.bsc.es/) and T2D knowledge portal (URL: www.type2diabetesgenetics.org), respectively, to examine whether or not there are T2D associated genetic variants located in the region encompassing +/-100 kb either upstream or downstream of EXOC6 and EXOC6B genes.
First of all, the URL: www.type2diabetesgenetics.org is outdated, and should be updated with the latest URL: https://t2d.hugeamp.org/
Second, based on the TIGER data portal, as presented in Table 1, the authors discovered 3 non-coding SNPs located in EXOC6 gene region, i.e., rs947591, rs2488071, and rs2488073, that reached genome-wide significance [i.e., P-value < 5*10^(-8)] for association with T2D only in "DIAGRAM DIAMANTE" and "DIAGRAM 1000G" respectively, but not in "70KforT2D". Based on the TIGER data portal, as presented in Table 2, for the 9 selected SNPs (rs529721336, rs557567903, rs776240131, rs79506623, rs142978146, rs186249148, rs2192015, rs138021417, and rs143622603) located in EXOC6B gene region none of them reached genome-wide significance [i.e., P-value < 5*10^(-8)] for association with T2D in "DIAGRAM DIAMANTE", "DIAGRAM 1000G", or "70KforT2D". but in "DIAGRAM DIAMANTE", the associations of rs529721336 (EXOC6B Non-Coding Variant), rs557567903 (EXOC6B Intron Variant), and rs776240131 (EXOC6B Intron Variant) were suggestively significant at a P-value threshold of P-value < 1*10^(-3). Then, based on T2D knowledge portal, EXOC6 gene region’s 3 non-coding SNPs, , i.e., rs947591, rs2488071, and rs2488073, reached genome-wide significance [i.e., P-value < 5*10^(-8)] for associations with T2D and with T2D adj BMI, respectively. There are 2 major deficiencies for this point:
(i) For the T2D knowledge portal, Table 3 should also include the EXOC6B gene region’s 3 non-coding SNPs discovered at P-value < 10^(-3) level as shown in Table 2 DIAGRAM DIAMANTE" subsection: rs529721336 (EXOC6B Non-Coding Variant), rs557567903 (EXOC6B Intron Variant), and rs776240131 (EXOC6B Intron Variant) for their potential associations with various T2D-related parameters in T2D knowledge portal, because although these non-coding variants did not reach genome-wide significance, they were located in EXOCB gene region,  which could be considered as suggestive T2D risk variants using P-value threshold of 10^(-3); and
(ii) The authors should have performed extensive molecular experiments to examine the potential functional effects of the 3 non-coding SNPs located in EXOC6 gene region: rs947591, rs2488071, and rs2488073, and the 3 non-coding SNPs located in EXOC6B gene region: rs529721336 (EXOC6B Non-Coding Variant), rs557567903 (EXOC6B Intron Variant), and rs776240131 (EXOC6B Intron Variant), and these could offer significant insights on how these genetic variations might affect the gene expressions of EXOC6 and EXOC6B, respectively, however, the authors only described several published studies on genetic associations of these SNPs with T2D as shown in "Discussion" section on Page 11, lines 325-336, but did not perform any original functional experiments to show how these non-coding SNPs might have a functional role in dysregulation of EXOC6 and EXOC6B gene expressions. These are also major limitations of this study. 

(3) Using INS-1 (832/13) cell line as an in vitro cell model, the authors showed that the silencing of EXOC6 and EXOC6B gene expressions could affect insulin secretion, insulin content, exocytosis machinery, glucose uptake. However, these are based on the assumption that EXOC6 and EXOC6B gene expressions are decreased in diabetic pancreatic islets, but their data presented in Section 3.1 based on the 2 publicly available microarray and RNA-seq data sets did not show that EXOC6 and EXOC6B gene expressions are down-regulated in diabetic pancreatic islets, and the authors themselves have no human or in vivo animal model data to show that EXOC6 and EXOC6B gene expressions are reduced in diabetic pancreatic islet tissues. This makes their in vitro cell model data are weak to address the point that EXOC6 and EXOC6B gene expressions are important in pathophysiology of T2D.

(4) The authors selected a set of pancreatic beta-cell function genes, i.e., Ins1, Ins2, Pdx1, NeuroD1, Gck, Glut2, InsR and Vamp2, to examine whether their mRNA expression levels (measured by qPCR) and protein expression levels (measured by Western blot analysis) could be changed by silencing of either Exoc6 gene or Exoc6B gene in the INS-1 (832/13) cell. These assays yielded mixed results, and by and large, the silencing of EXOC6 and EXOC6B gene expressions did not lead to consistent down-regulations of these pancreatic beta-cell function genes, and it is puzzling why for at mRNA level, Gck gene expression was down-regulated in Exoc6b-silenced cells but at protein level, Gck gene expression was not affected, and at mRNA level, InsR gene expression was up-regulated in Exoc6b-silenced cells but at protein level, InsR gene expression was down-regulated, but did not reach statistical significance.

(5) Page 11, lines 339-342, the authors stated that
"Gene si-lencing of Exoc6/6b in INS-1 cells impaired insulin secretion reduced insulin content and decreased glucose uptake without affecting cell viability or cytotoxic apoptosis levels (Fig-ure X).".
The authors shall correct the Figure X to correct figure numbers.

Taken together, there are major deficiencies and significant concerns for this manuscript. The most important concern is that the microarray and RNA-seq gene expression data sets did not support that the expressions of EXOC6 and EXOC6B were down-regulated in pancreatic islets in diabetic state compared to non-diabetic state. Although the authors identified several genome-wide significant non-coding SNPs in EXOC6 gene (rs947591, rs2488071, and rs2488073) and suggestively significant non-coding SNPs in EXOC6B gene (rs529721336, rs557567903, and rs776240131) by using publicly available TIGER data portal (URL: http://tiger.bsc.es/)and T2D knowledge portal, such genetic data were not original data generated by the authors or the authors did extensive statistical analysis on original human studies’ data, but rather directly extracted such summary statistics from online data portals, which also limited the originality of this paper. Also, the bulk of the paper did not try to disentangle how the 3 non-coding genetic variants located in EXOC6 gene and 3 non-coding genetic variants located in EXOC6B gene could impact the gene expressions using original functional experiments. The siRNA experiments conducted by the authors to examine how dysregulations of EXOC6 gene and EXOC6B gene may affect expressions of several pancreatic beta-function genes in an in vitro cell model are weak, and there are discrepancies between the qPCR results and Western blot analysis results. Overall, the data lack originality and although it appears that a set of molecular assays were performed, they were all based on the assumption that EXOC6 and EXOC6B gene expressions in pancreatic islet were down-regulated in diabetic state, which are unproven by the gene expression data sets shown in this manuscript, and there are discrepancies of the in vitro experiments at RNA level and at protein level, which are difficult to interpret.

(II) Minor Comments

There are many typographical, grammatical errors and inadequacies contained in this manuscript. For example, In the main text (i.e., "Simple Summary", "Abstract", "1. Introduction", "2. Materials and Methods", "3. Results", and "4. Discussion"), all occurrences of "downregulated" should be corrected to "down-regulated", and all occurrences of downregulation" should be corrected to "down-regulation", and also, "upregulated" should be corrected to "up-regulated", and all occurrences of upregulation" should be corrected to "up-regulation", respectively.

In the following, only a very few examples of corrections for typographical and grammatical errors are provided for the authors to show that the authors should always perform a thorough English checking and editing to correct all typographical and grammatical errors before manuscript submission, which is only a basic requirement for submitting a scientifically sound and well-written manuscript to a peer-reviewed scientific journal (improvement of English editing is required BUT NOT adequate to substantially improve the scientific contents of a manuscript, and scientific contents must be dramatically improved, which are the major determining factor for the quality of a manuscript, in conjunction with improvements in English writing): 

Page 1, line 18,
"demonstrated that EXOC/6B"
could be corrected to
"demonstrated that EXOC6/6B"

Page 1, lines 25-26,
"microarray and RNA-sequence expression data"
could be corrected to
"microarray and RNA-seq expression data"

Page 1, line 26,
"expression of EXOC/6B in"
could be corrected to
"expression of EXOC6/6B in"

Page 2, line 80,
"Microarray and RNA-sequencing data"
could be corrected to
Microarray and RNA-sequencing (RNA-seq) data"

Page 2, line 90,
"GEO publicly available database (accession number; GSE50398 and GSE41762)[28,29]"
could be corrected to
"National Center for Biotechnology Information (NCBI) Gene Expression Omnibus (GEO) publicly available database (accession numbers: GSE50398 and GSE41762) [28,29]"
As shown in above, there shall always be a blank space placed before the square brackets for citing reference(s) in the main text.

Page 3, line 99,
"provided from Dr. Newgard"
could be corrected to
"provided from Dr. C. Newgard"
This above correction is based on the following PLoS One 2013 paper:
Hectors TL, Vanparys C, Pereira-Fernandes A, et al.,Evaluation of the INS-1 832/13 cell line as a beta-cell based screening system to assess pollutant effects on beta-cell function. PLoS One. 2013;8:e60030. PMID: 23555872

Page 8, line 242,
"and Exoc/6b as determined"
could be corrected to
"and Exoc6b as determined"
In above, "Exoc6b" could be in italic font.

The above are just very few examples (i.e., a very small portion of all corrections that should be made) for correcting the entire manuscript.

Author Response

Reviewer 2

treatment of T2D.

(I) Major Comments

I have significant concerns that are elaborated in the following.

(1) As indicated in Section 2.1, the authors used 2 Microarray and RNA-sequencing (RNA-seq) data sets from human pancreatic islets obtained from NCBI Gene Expression Omnibus (GEO) database, i.e., GSE50398 [URL: https://www.ncbi.nlm.nih.gov/geo/query/acc.cgi?acc=GSE50398; Platforms: expression profiling by Affymetrix Human Gene 1.0 ST Array and Illumina HiSeq 2000 (Homo sapiens)] and GSE41762 (URL: https://www.ncbi.nlm.nih.gov/geo/query/acc.cgi?acc=GSE41762; Platform: Affymetrix Human Gene 1.0 ST Array), respectively, but the gene expression levels for EXOC6 and EXOC6B genes were not changed in diabetic pancreatic islets compared to non-diabetic pancreatic islets. These RNA gene expression results are very essential, but they do not provide support for potential dysregulations of EXOC6 and EXOC6B genes at the mRNA level between diabetic and non-diabetic pancreatic islet tissues. Further, these gene expression data sets are not original data set generated by the authors, but were publicly available datasets stored in NCBI GEO database, and this severely limited the originality of the study, and the fact that EXOC6 and EXOC6B genes were not differentially expressed in 2 independent microarray and RNA-seq data sets were essentially negative findings, which did not support the authors’ study hypothesis, which is another major limitation of this study.

Answer: We thank the reviewer for the comment, which is very important. Our hypothesis for the present study is that EXOC6/6B are crucial regulators for insulin secretion in β-cells. The observation that the gene expression levels for EXOC6 and EXOC6B genes were not changed in diabetic pancreatic islets compared to non-diabetic pancreatic islets reflects the fact that the majority of these diabetic donors suffer from insulin resistance rather than insulin secretion or could be a compensatory response that regulates the expression of EXOC6/6B. In the revised manuscript, we highlighted this issue;

 “A possible explanation could be a compensatory response that regulates the expression of EXOC6/6B in β-cell or reflects the fact that the diabetic donors suffer from insulin resistance rather than insulin secretion.”

Regarding the gene expression dataset, we did generate these data (see reference below),

  • Identification of novel genes for glucose metabolism based upon expression pattern in human islets and effect on insulin secretion and glycemia. J Taneera, J Fadista, E Ahlqvist, D Atac, E Ottosson-Laakso, CB Wollheim, et al. Human molecular genetics 24 (7), 1945-1955.
  • Global genomic and transcriptomic analysis of human pancreatic islets reveals novel genes influencing glucose metabolism. J Fadista, P Vikman, EO Laakso, IG Mollet, JL Esguerra, J Taneera et al. Proceedings of the National Academy of Sciences 111 (38), 13924-13929

(2) The authors searched 2 online data portals, i.e., TIGER data portal (URL: http://tiger.bsc.es/) and T2D knowledge portal (URL: www.type2diabetesgenetics.org), respectively, to examine whether or not there are T2D associated genetic variants located in the region encompassing +/-100 kb either upstream or downstream of EXOC6 and EXOC6B genes.
First of all, the URL: www.type2diabetesgenetics.org is outdated, and should be updated with the latest URL: https://t2d.hugeamp.org/
Second, based on the TIGER data portal, as presented in Table 1, the authors discovered 3 non-coding SNPs located in EXOC6 gene region, i.e., rs947591, rs2488071, and rs2488073, that reached genome-wide significance [i.e., P-value < 5*10^(-8)] for association with T2D only in "DIAGRAM DIAMANTE" and "DIAGRAM 1000G" respectively, but not in "70KforT2D". Based on the TIGER data portal, as presented in Table 2, for the 9 selected SNPs (rs529721336, rs557567903, rs776240131, rs79506623, rs142978146, rs186249148, rs2192015, rs138021417, and rs143622603) located in EXOC6B gene region none of them reached genome-wide significance [i.e., P-value < 5*10^(-8)] for association with T2D in "DIAGRAM DIAMANTE", "DIAGRAM 1000G", or "70KforT2D". but in "DIAGRAM DIAMANTE", the associations of rs529721336 (EXOC6B Non-Coding Variant), rs557567903 (EXOC6B Intron Variant), and rs776240131 (EXOC6B Intron Variant) were suggestively significant at a P-value threshold of P-value < 1*10^(-3). Then, based on T2D knowledge portal, EXOC6 gene region’s 3 non-coding SNPs, , i.e., rs947591, rs2488071, and rs2488073, reached genome-wide significance [i.e., P-value < 5*10^(-8)] for associations with T2D and with T2D adj BMI, respectively. There are 2 major deficiencies for this point:

(i) For the T2D knowledge portal, Table 3 should also include the EXOC6B gene region’s 3 non-coding SNPs discovered at P-value < 10^(-3) level as shown in Table 2 DIAGRAM DIAMANTE" subsection: rs529721336 (EXOC6B Non-Coding Variant), rs557567903 (EXOC6B Intron Variant), and rs776240131 (EXOC6B Intron Variant) for their potential associations with various T2D-related parameters in T2D knowledge portal, because although these non-coding variants did not reach genome-wide significance, they were located in EXOCB gene region,  which could be considered as suggestive T2D risk variants using P-value threshold of 10^(-3); and

Answer: We thank the reviewer for the comment.  In the revised manuscript, we omitted a large portion of the genetics data (including T2D knowledge portal) as suggested by the other reviewer. We only showed the most robust data (see the table below).

 Table 2: SNPs in the proximity of EXOC6 (± 100 Kb) that associated with T2D in TIGER data portal

DIAGRAM DIAMANTE

ID 

Reference allele 

Alternate allele 

Type 

Sample size 

Effect Allele 

MAF 

OR 

SE 

p-value 

rs947591

C

A

Upstream

231420

A

0.48

1.07

0.0064

2.4E-25

rs2488071

A

G

Upstream

231420

A

0.46

1.048

0.0064

9.1E-21

rs2488073

A

G

Upstream

231420

A

0.48

1.047

0.0064

2.5E-20

DIAGRAM 1000G

rs947591

C

A

Upstream

A

-

1.09

0.013

7.4E-13

rs2488073

A

G

Upstream

A

-

0.917

0.013

8.5E-12

rs2488071

A

G

Upstream

A

-

0.919

0.013

3.1E-11

(ii) The authors should have performed extensive molecular experiments to examine the potential functional effects of the 3 non-coding SNPs located in EXOC6 gene region: rs947591, rs2488071, and rs2488073, and the 3 non-coding SNPs located in EXOC6B gene region: rs529721336 (EXOC6B Non-Coding Variant), rs557567903 (EXOC6B Intron Variant), and rs776240131 (EXOC6B Intron Variant), and these could offer significant insights on how these genetic variations might affect the gene expressions of EXOC6 and EXOC6B, respectively, however, the authors only described several published studies on genetic associations of these SNPs with T2D as shown in "Discussion" section on Page 11, lines 325-336, but did not perform any original functional experiments to show how these non-coding SNPs might have a functional role in dysregulation of EXOC6 and EXOC6B gene expressions. These are also major limitations of this study. 

Answer: This is very important point. As we stated in the result secretion, we investigated the effect of these SNPs (rs947591, rs2488071, and rs2488073) on the expression of EXOC6 and EXOC6B in human pancreatic islets using TIGER portal (aggregating more than 400 human islet genomic datasets from five cohorts) as a mean of eQTL. Data in TIGER portal are imputed genotypes using four reference panels and meta-analyze cohorts to improve the coverage of expression quantitative trait loci (eQTL). However, we could not see any effect of rs947591, rs2488071or rs2488073 on the expression of EXOC6 or EXOC6B in human pancreatic islets. Still the eQTLs on other tissues such as blood or muscle warrant further investigations.   

A New statement has been added to the revised manuscript to clarify this point. “Importantly, exploring the effect of rs947591, rs2488071, and rs2488073 on EXOC6 and EXOC6B expression in human pancreatic islets using TIGER portal revealed no effect. Further investigations are needed to explore the effect of the three identified SNPs on other tissues such muscle or blood.”   

(3) Using INS-1 (832/13) cell line as an in vitro cell model, the authors showed that the silencing of EXOC6 and EXOC6B gene expressions could affect insulin secretion, insulin content, exocytosis machinery, glucose uptake. However, these are based on the assumption that EXOC6 and EXOC6B gene expressions are decreased in diabetic pancreatic islets, but their data presented in Section 3.1 based on the 2 publicly available microarray and RNA-seq data sets did not show that EXOC6 and EXOC6B gene expressions are down-regulated in diabetic pancreatic islets, and the authors themselves have no human or in vivo animal model data to show that EXOC6 and EXOC6B gene expressions are reduced in diabetic pancreatic islet tissues. This makes their in vitro cell model data are weak to address the point that EXOC6 and EXOC6B gene expressions are important in pathophysiology of T2D.

Answer: The main rationale for studies EXOC6/6B genes involvement in the pathophysiology of T2D was our recent proposed finding that the EXOC6/6B gene might be an important molecular link between dysfunctional pancreatic islets and ciliated lung epithelium that makes diabetic patients more susceptible to severe SARS-COV-2 complications (Cellular exocytosis gene (EXOC6/6B): a potential molecular link for the susceptibility and mortality of COVID-19 in diabetic patients. Hachim, IY Hachim, S Al Heialy, J Taneera. bioRxiv). We did not build up our hypotheses based on the assumption that EXOC6/6b are downregulated in human islets. In fact, we used the gene expression data to provide more evidence for EXOC6/6B in pancreatic beta cells but not the main rationale.

Even though EXOC6/6B are not downregulated in human islets, we believe that we present novel finding for the role of EXOC6/6B in pancreatic beta cell function and its established role in the secretory granule docking.

(4) The authors selected a set of pancreatic beta-cell function genes, i.e., Ins1, Ins2, Pdx1, NeuroD1, Gck, Glut2, InsR and Vamp2, to examine whether their mRNA expression levels (measured by qPCR) and protein expression levels (measured by Western blot analysis) could be changed by silencing of either Exoc6 gene or Exoc6B gene in the INS-1 (832/13) cell. These assays yielded mixed results, and by and large, the silencing of EXOC6 and EXOC6B gene expressions did not lead to consistent down-regulations of these pancreatic beta-cell function genes. It is puzzling why for at mRNA level, Gck gene expression was down-regulated in Exoc6b-silenced cells but at protein level, Gck gene expression was not affected. At mRNA level, InsR gene expression was up-regulated in Exoc6b-silenced cells but at protein level, InsR gene expression was down-regulated, but did not reach statistical significance.

Answer: We agree with the comment. As noticed, the transcriptional changes of GCK and InsR were not mirrored by protein expression. This could be explained by the post-transcriptional regulations that might affect mRNA stability and translation rate.

(5) Page 11, lines 339-342, the authors stated that
"Gene silencing of Exoc6/6b in INS-1 cells impaired insulin secretion reduced insulin content and decreased glucose uptake without affecting cell viability or cytotoxic apoptosis levels (Figure X).".
The authors shall correct the Figure X to correct figure numbers.

Answer: Thanks, Corrected.

Taken together, there are major deficiencies and significant concerns for this manuscript. The most important concern is that the microarray and RNA-seq gene expression data sets did not support that the expressions of EXOC6 and EXOC6B were down-regulated in pancreatic islets in diabetic state compared to non-diabetic state. Although the authors identified several genome-wide significant non-coding SNPs in EXOC6 gene (rs947591, rs2488071, and rs2488073) and suggestively significant non-coding SNPs in EXOC6B gene (rs529721336, rs557567903, and rs776240131) by using publicly available TIGER data portal (URL: http://tiger.bsc.es/)and T2D knowledge portal, such genetic data were not original data generated by the authors or the authors did extensive statistical analysis on original human studies’ data, but rather directly extracted such summary statistics from online data portals, which also limited the originality of this paper. Also, the bulk of the paper did not try to disentangle how the 3 non-coding genetic variants located in EXOC6 gene and 3 non-coding genetic variants located in EXOC6B gene could impact the gene expressions using original functional experiments. The siRNA experiments conducted by the authors to examine how dysregulations of EXOC6 gene and EXOC6B gene may affect expressions of several pancreatic beta-function genes in an in vitro cell model are weak, and there are discrepancies between the qPCR results and Western blot analysis results. Overall, the data lack originality and although it appears that a set of molecular assays were performed, they were all based on the assumption that EXOC6 and EXOC6B gene expressions in pancreatic islet were down-regulated in diabetic state, which are unproven by the gene expression data sets shown in this manuscript, and there are discrepancies of the in vitro experiments at RNA level and at protein level, which are difficult to interpret.

Answer: Please see our respond above to the raised comments point by point.

(II) Minor Comments

There are many typographical, grammatical errors and inadequacies contained in this manuscript. For example, In the main text (i.e., "Simple Summary", "Abstract", "1. Introduction", "2. Materials and Methods", "3. Results", and "4. Discussion"), all occurrences of "downregulated" should be corrected to "down-regulated", and all occurrences of downregulation" should be corrected to "down-regulation", and also, "upregulated" should be corrected to "up-regulated", and all occurrences of upregulation" should be corrected to "up-regulation", respectively.

Answer: Thanks. Corrected.

In the following, only a very few examples of corrections for typographical and grammatical errors are provided for the authors to show that the authors should always perform a thorough English checking and editing to correct all typographical and grammatical errors before manuscript submission, which is only a basic requirement for submitting a scientifically sound and well-written manuscript to a peer-reviewed scientific journal (improvement of English editing is required BUT NOT adequate to substantially improve the scientific contents of a manuscript, and scientific contents must be dramatically improved, which are the major determining factor for the quality of a manuscript, in conjunction with improvements in English writing): 

Page 1, line 18,
"demonstrated that EXOC/6B"
could be corrected to
"demonstrated that EXOC6/6B"

Corrected.

Page 1, lines 25-26,
"microarray and RNA-sequence expression data"
could be corrected to
"microarray and RNA-seq expression data"

Corrected.

Page 1, line 26,
"expression of EXOC/6B in"
could be corrected to
"expression of EXOC6/6B in"

Corrected.

Page 2, line 80,
"Microarray and RNA-sequencing data"
could be corrected to
Microarray and RNA-sequencing (RNA-seq) data"

Corrected.

Page 2, line 90,
"GEO publicly available database (accession number; GSE50398 and GSE41762)[28,29]"
could be corrected to "National Center for Biotechnology Information (NCBI) Gene Expression Omnibus (GEO) publicly available database (accession numbers: GSE50398 and GSE41762) [28,29]"
As shown in above, there shall always be a blank space placed before the square brackets for citing reference(s) in the main text.

Corrected.

Page 3, line 99,
"provided from Dr. Newgard"
could be corrected to
"provided from Dr. C. Newgard"
This above correction is based on the following PLoS One 2013 paper:
Hectors TL, Vanparys C, Pereira-Fernandes A, et al.,Evaluation of the INS-1 832/13 cell line as a beta-cell based screening system to assess pollutant effects on beta-cell function. PLoS One. 2013;8:e60030. PMID: 23555872.

Corrected. Reference was added.

Page 8, line 242,
"and Exoc/6b as determined"
could be corrected to
"and Exoc6b as determined"
In above, "Exoc6b" could be in italic font.

Corrected.

The above are just very few examples (i.e., a very small portion of all corrections that should be made) for correcting the entire manuscript.

Answer: Thanks a lot.

Reviewer 3 Report

Paper by  Sulaiman et al. present some interesting data on the role of one exocyst complex component on insulin secretion by Beta-islet in vitro. However some critical points necessitate to be ameliorate.

1. SNP exploration by tiger data portal might  not add data to paper per se. There are a very limited number of population studies to consider results of this theoretical approach evaluable. It appear necessary that authors study the association with TD2 on a adequate sample of really recruited patients.

As an alternative paper might be reshaped eliminating SNP studies

2. Silencing experiments suggest that Exoc6/Exoc6-b play a central role in modulation of genes implied in insulin production and secretion processes. However as the  exocytosis machinery involves other  exocyst complex components the effect of silencing of these subunit genes apear essential to better define Exoc6/Exoc6-b role in insulin related gene pathways.

minor:

paper should be checked for some typo errors

Author Response

Reviewer 3

Paper by  Sulaiman et al. present some interesting data on the role of one exocyst complex component on insulin secretion by Beta-islet in vitro. However some critical points necessitate to be ameliorate.

  1. SNP exploration by tiger data portal might not add data to paper per se. There are a very limited number of population studies to consider results of this theoretical approach evaluable. It appear necessary that authors study the association with TD2 on a adequate sample of really recruited patients. As an alternative paper might be reshaped eliminating SNP studies.

Answer: We totally agree with the comment. In fact, we have already started recruiting DNA samples to study the association of the top three SNPs in DIAGRAM Diamante (rs947591, rs2488071 and rs2488073) and T2D Emirati population. The rationale for adding these data in the paper is providing more evidence to support the association with T2D side-by-side to the functional data. As suggested by the reviewer, we omitted some of these data (Table 2 and Table 3) in the revised manuscript and left only the most robust data (Table 1). 

  1. Silencing experiments suggest that Exoc6/Exoc6-b play a central role in modulation of genes implied in insulin production and secretion processes. However as the exocytosis machinery involves other exocyst complex components the effect of silencing of these subunit genes apear essential to better define Exoc6/Exoc6-b role in insulin related gene pathways.

Answer: This is a valid point. The effect of other subunits on pancreatic beta-cell function is essential for better understanding as different subunits could play an important role in beta cell biology. Unfortunately, due to limited resources and time, we cannot investigate this in the current study. It will be very interesting to look it up as a follow-up study. However, in the revised version, we investigated whether silencing of Exoc6 or Exoc6b in INS-1 cells has any impact on other subunits' expression. As shown in the figure below, only Exoc5 was shown to be affected by Exoc6 silencing. These data were added to the revised manuscripts. 

 minor:

paper should be checked for some typo errors

Answer: Done.

Round 2

Reviewer 1 Report

I am satisfied with the corrections brought by the authors, and recommend acceptance in the present form

Author Response

Thanks

Reviewer 2 Report

In this revision 1 (i.e., R1) manuscript, the authors have addressed major and minor comments of the reviewers. I have the following further comments.
First of all, all occurrences of "SARS-COV-2" (on line 35, line 86, line 343 of the manuscript) should be corrected to "SARS-CoV-2" because "SARS-CoV-2" is the standard term to be used.
Further, the following typographical and grammatical errors should be corrected:

(1) Page 3, line 54
"T2D accounts"
could be corrected to
"Type 2 diabetes (T2D) accounts"

(2) Page 3, lines 59-60,
"Classically, elevated ATP concentration closes the sensitive K+-chan-nels in response to glucose stimulation [6] and opens the voltage-gated Ca2+ channels to permit Ca2+ influx and trigger the exocytosis of insulin granules."
In above statement the "+" in "K+" and the 2 occurrences of "2+" in "Ca2+" shall be in superscript font.

(3) Page 3, line 73,
"was reported to be defected"
could be corrected to
"was reported to be affected"

(4) Page 3, line 84,
"EXOC6/6B gene might be"
could be corrected to
"EXOC6/6B genes might be"
This is because EXOC6 and EXOC6B are 2 different human genes, not a single gene.

(5) Page 3, line 89,
"Microarray and RNA-sequencing data"
could be corrected to
"Microarray and RNA-sequencing (RNA-seq) data"

(6) Page 3, line 91,
"Using the TIGER portal,"
could be corrected to
"Using the translational human pancreatic islet genotype tissue-expression resource (TIGER) portal"

(7) Page 4, line 116
"Exoc6 (s132217) or Exoc6-b"
could be corrected to
"Exoc6 (s132217) or Exoc6b"

(8) Page 4, line 119,
"secretion assay buffer (SAB)[35]"
could be corrected to
"secretion assay buffer (SAB) [35]"
The authors shall ensure that in the main text, there is a single blank space before citations of references.

(9) Page 5, line 163,
"2.9. Quantitative-PCR"
could be corrected to
"2.9. Quantitative PCR (qPCR)"

(10) Page 5, line 172,
"2−ΔΔCT method"
The "−ΔΔCT" should be in superscript font

(11) Page 6, lines 200-201,
"student t-test or nonparametric Mann-Whitney tests"
could be corrected to
"Student's t-test or nonparametric Mann-Whitney U tests"

(12) Page 7, line 250,
"As shown in figure 2C-D,"
could be corrected to
"As shown in Fig. 2C-D,"
In above, "Fig. 2C-D" shall be in italic font

(13) Page 7, line 259,
"figure 2D-E
could be corrected to
"Fig. 2D-E"
In above, "Fig. 2D-E" shall be in italic font

(14) Page 11, line 341,
"defect in the exocytosis"
could be corrected to
"defects in the exocytosis"

(15) Page 11, lines 372-373,
"apoptosis levels (Fig-ure 2 and 3)"
could be corrected to
"apoptosis levels (Figs. 2 and 3)"

(16) Page 12, line 391,
"might defect the insulin secretion"
could be corrected to
"might affect the insulin secretion"

Author Response

We thank the reviewer for careful reading and suggestions.

As suggested we revised all raised comments.

Reviewer 3 Report

Sulaiman et al. have improved the quality of paper and answered with efficacy to all criticisms and  suggestions formulated on the previous version of the manuscript. I think that paper would be suitable for publication on Biology.

Author Response

Thanks